# Interactions between Nanoparticles and Intestine

**DOI:** 10.3390/ijms23084339

**Published:** 2022-04-14

**Authors:** Manuela Vitulo, Elisa Gnodi, Raffaella Meneveri, Donatella Barisani

**Affiliations:** School of Medicine and Surgery, University of Milano-Bicocca, 20900 Monza, Italy; m.vitulo1@campus.unimib.it (M.V.); e.gnodi@campus.unimib.it (E.G.); raffaella.meneveri@unimib.it (R.M.)

**Keywords:** nanoparticles, nanocarriers, insulin delivery, inflammatory bowel diseases, colon cancer, food additives

## Abstract

The use of nanoparticles (NPs) has surely grown in recent years due to their versatility, with a spectrum of applications that range from nanomedicine to the food industry. Recent research focuses on the development of NPs for the oral administration route rather than the intravenous one, placing the interactions between NPs and the intestine at the centre of the attention. This allows the NPs functionalization to exploit the different characteristics of the digestive tract, such as the different pH, the intestinal mucus layer, or the intestinal absorption capacity. On the other hand, these same characteristics can represent a problem for their complexity, also considering the potential interactions with the food matrix or the microbiota. This review intends to give a comprehensive look into three main branches of NPs delivery through the oral route: the functionalization of NPs drug carriers for systemic targets, with the case of insulin carriers as an example; NPs for the delivery of drugs locally active in the intestine, for the treatment of inflammatory bowel diseases and colon cancer; finally, the potential concerns and side effects of the accidental and uncontrolled exposure to NPs employed as food additives, with focus on E171 (titanium dioxide) and E174 (silver NPs).

## 1. Introduction

The development of nanotechnology in recent years has dramatically changed the approaches for drug delivery, initially improving the efficacy of the intravenous route for various drugs, such as anti-cancer ones. These results have also prompted researchers and pharma companies to look into the possibility of using nanocarriers for drug delivery through the oral route. On the other hand, in addition to pharmacological use, nanoparticles (NPs) have also been used quite extensively in the food industry, due to their ability to improve food characteristics, as well as product shelf-life. This implies that the interactions between NPs and the intestine can become quite frequent; for this reason, it is necessary to understand the pros and cons of the NPs presence in the intestine, also considering the possible interactions with the lumen components (food, acid environment, enzymes) and the different cell types. This review will focus on three different uses of the NPs, i.e., vehicle for systemic drugs, delivery system for drugs acting on the intestine and food additives. The interest in this field is in fact growing, as demonstrated by the increasing number of studies published in the literature. In particular, the interest about a potential administration through the oral route has gained much attention, as shown in Figure 1.

First of all, general aspects regarding the intestine, the NPs and their interaction needs to be elucidated, to better understand the broad applications, possible functionalization, and concerns that will be presented in this work.

### 1.1. The Intestinal Barrier

The gastrointestinal tract is a difficult environment for nanocarriers, both due to the aggressive conditions present in the lumen, as well as for the presence of the barrier separating the lumen from the rest of the body. The first challenge is a large pH gradient, ranging from pH 1–2.5 in the stomach to pH 7–8 in the colon, fact which can affect the structure of the nanocarriers or of the vehiculated drug. Moreover, the lumen enzymes, both in the stomach (ex. Pepsin) and in the duodenum (biliary and pancreatic secretions that include lipases, peptidases, and amylases) can affect the nanocarrier stability and/or their capacity to bind different substances (including food components) [1,2]. 

The intestine is composed of different cell types that have specific functions, and the composition changes according to the anatomical site, i.e., the small or large intestine. In the small intestine the main absorption function is performed by the enterocytes, which are also responsible for the tight junctions, the most important structure creating the intestinal barrier. The mechanisms of uptake of lumen substances could be either paracellular or transcellular, i.e., through the enterocytes. The paracellular pathway usually plays a minor role in the passage of NPs, which are usually transported through the transcellular route. This occurs through vesicle-mediated mechanisms, either endocytosis or pinocytosis; it is easy to understand that the intrinsic characteristics of the NPs can affect the ability to bind to the enterocytes and to be transported through the transcellular route. In Figure 2, the main NPs administration routes are represented, along with the main aspects related to the oral administration (Figure 2).

There is also another point that needs to be considered, i.e., the presence of multidrug resistance transporters (MDR) in the epithelium, fact which could dampen the total amount of the drug bound to NPs which had been taken up by the enterocytes. In addition to enterocytes, there are other cells in the small intestine, such as goblet cells (localized in the villi), as well as Paneth and stem cells (in the crypts); the former are producing the mucus which covers the intestinal epithelium, whereas Paneth cells are responsible for the production of antimicrobial peptides and immunomodulating proteins. Mucus is a complex hydrogel composed of water and different types of proteins, among which mucins are the most abundant ones. Most mucins are glycosylated, so they have a negative charge, characteristic which could lead to the adhesion of positively charged nanocarriers through electrostatic interactions. This ability of nanocarriers to bind to the mucus layer could be regarded only partially as positive, since the intestinal mucus is structured in two different layers: the first one, nearer to the intestinal lumen, is more loose whereas the layer in contact with the epithelium is firmly adhered. The firm binding of the NPs to the upper layer can, thus, lead to a prompt clearance and to a reduction in the opportunity to reach the epithelium [3].

It must also be mentioned that the intestine represents the point of entry of several foreign antigens, and the immune system present in this organ is responsible for inducing tolerance or an immune response. In this regard, M cells play an important role in the intestinal immune system, which also includes gut-associated lymphoid tissues and mesenteric lymph nodes. M cells are a specialized type of intestinal epithelial cells lacking microvilli and the mucus layer on its surface, essential for sampling the intestinal content and vehiculate the lumen antigens to the antigen presenting cells (APCs) present in the sub-epithelial region. Within APCs, antigens are processed and presented to CD4+ T cells to induce T and B cell responses in the lymphocytes that are dispersed in the epithelium (intraepithelial lymphocytes) and in the lamina propria. The presence of the immune system in the intestine is not irrelevant, considering that ingested NPs (either vehiculating drugs or ingested with food) could interact with all these components, either being altered in their composition or modifying the immune response. In fact, the interaction with the immune system can be regarded as a double-edge sword, since it could be useful for the possible development of NPs-based oral vaccines but, on the other hand, could trigger an allergic/autoimmune response against normal intestinal components [4,5,6]. Last but not least, there are two other interactions that must be kept in mind; the first one is related to the absorption route, and, in particular, the possibility that the NPs and their load are removed from the circulation during the first hepatic passage, whereas the second regards an important component of the gut lumen, i.e., the microbiota. As concerns the absorption route, it must be remembered that NPs can be absorbed either through the blood flow or through the lymphatic system; this difference is quite important, since blood coming from the intestine is drained by the portal vein, and the passage through the liver can, in fact, affect the fate of the nanocarrier due to the resident macrophage uptake. As recently reviewed by Zhang et al., the lymphatic system could actually be employed for drug delivery, but in order to achieve this goal, drugs should use the chylomicrons path (through enterocytes) or M cells and Peyer’s patches [4]. In the first case, the lipophilicity of the drug/carrier is a necessary condition, whereas the passage through M cells can be increased by the use of specific peptides (see below). 

Regarding the interaction with the microbiota, the published papers have mainly focused on the effect on the colonic microbiota, showing variations in its composition after NPs exposure. However, the data collected up to now are partial and, in most cases, providing information regarding variations at the phyla level (i.e., of categories including a large number of different species [7,8,9,10,11]). The human gut contains about 1000 different species of bacteria, and 50 phyla. In the latter category, four phyla are the most abundant: *Bacteroidetes* (Gram-negative bacteria), *Firmicutes* (Gram-positive, aerobic, and anaerobic bacteria), *Proteobacteria* (such as *Escherichia*), and *Actinobacteria* (for example *Bifidobacterium*). Of these, the predominant phyla, accounting for up to 70 to 90% of total bacteria, are the *Firmicutes* (60–80%) and the *Bacteroidetes* (20–30%) [12]. The gut microbiota plays important roles in a number of physiological functions, such as the digestion of dietary fiber or the production of key metabolites for the host. However, it is also involved in the maintenance of structural integrity of the mucosal barrier, in the regulation of the immune response and the protection against pathogens [12]. Variations in the Firmicutes/Bacteroidetes phyla ratio have been reported in several disorders, such as obesity, inflammatory bowel diseases (IBD), celiac disease (CeD) etc. [13,14,15]. It still remains to be determined if the alteration of the microbiota represents the real initiating factor of these disorders, although data obtained in mice showed the presence of intestinal damage in absence of microbial colonization of the intestine. Due to this pivotal role of the microbiota, an alteration of its composition induced by NPs exposure could have a severe effect on the general well-being of the individual [15]. 

Apart from being used as food additives to improve food aspect or taste, metallic nanoparticles (MNPs) and, in particular, silver (Ag) NPs have a strong potential as antimicrobial agents. In fact, foodborne diseases, mainly due to pathogenic bacteria genera *Salmonella*, *Listeria*, and *Escherichia*, are a risk for consumer health, especially for people in developing countries [16]. Therefore, the use of NPs as antimicrobial agents has been exploited, for example in filters for water treatment and in food/food packaging to improve food safety and its shelf-life. At the same time, NPs could be used in animal feeding, tackling the problem of antibiotic-resistance caused by the massive use of antibiotics in livestock [17]. The MNPs antibacterial activity is due to the interaction with the cell membrane, but mainly to the release of ions in the cells, which causes redox imbalances due to reactive oxygen species (ROS) production and the consequent cell death [16]. Despite the proved efficacy in counteracting pathogens, only a few studies focused on the impact on human microbiota and obtained contrasting results. In this regard, it has to be considered that in vivo models can be very different from the human microbiota composition. In fact, even if mouse and human microbiota are quite similar at phylum level, they are different in genera and species, fact that, in combination with the broad spectrum of the used NPs (different shape, dimension, core material, functionalization) and the different digestion, can explain the various results observed by researchers [18]. It has been reported, however, that the Firmicutes/Bacteroidetes phylum ratio can be modified by the exposure to AgNPs in in vitro models that employed human faecal samples [19,20]. Furthermore, modifications in bacterial cell vitality, number and populations were found in samples treated with Zinc oxide (ZnO) NPs, but these modifications differed in children with autism compared to healthy controls [8], suggesting that NPs effect on microbiota can be also influenced by the host condition or by the eventual dysbiosis already present [21]. Finally, it has been shown that the exposure to NPs can alter the microbiota composition, potentially decreasing its ability to counteract pathogens and creating a bacterial profile similar to the one found IBD, as recently reviewed by Ghebretatios et al. [11]. More information on TiO_2_ and Ag NPs are reported in the Section 5.

As regards the possible effect of other categories of NPs on microbiota composition, data are limited since most of the studies that evaluated this aspect were performed on functionalized NPs, either carrying a specific compound or targeting a particular condition [22,23,24,25]. However, some researchers focused on the effects of the “pure” NPs, showing alterations in the microbiota composition. Chandrarathna et al. observed an increase in Bacteroidetes abundance and a reduction in Firmicutes after the oral administration of pectin NPs [26], whereas Yu et al. detected variations in microbiota composition after a 7-day treatment with mesoporous silica NPs [27]. Variations in microbiota, but also in plasma metabolome were detected by Landsiedel et al., who treated rats by oral gavage of silicon dioxide (SiO_2_) or Ag NPs for 28 days, thus mimicking a “chronic” exposure [28]. Obviously more studies are needed, since additional factors, such as gender, could determine a variation in the effect of NPs on the microbiota, as recently suggested by Bredeck et al. [29]. Last but not least, a change in microbiota could generate a variation in the produced metabolites which, in turn, could alter the host physiology, as recently suggested by Diao et al. who detected neurobehavioral impairment in mice treated with SiO_2_ NPs [30]. Interestingly, the effect of NPs on microbiota does not necessarily require oral ingestion, since Chaplin et al. demonstrated an alteration in the gut microbiota after the systemic administration of poly (lactic-coglycolic acid) (PLGA) NPs, data that were explained by the ability of the injected NPs to reach the intestine through bile secretion [31]. Although these data were obtained in mice, they would suggest that the interaction NPs-microbiota could be more complex than originally expected.

### 1.2. Nanoparticles 

The NPs that can interact with the intestine can be divided into different categories, mainly according to the material used to generate them. 

Lipid-based nanocarriers have been quite extensively used in drug delivery because of their versatility, biocompatibility and low toxicity profile, and their use by i.v. administration has already been approved by Food and Drug Administration (FDA) and European Medicines Agency (EMA) [32] (recently reviewed by Halwani). However, the oral route presents a series of advantages, e.g., ease of administration and high patient compliance, and, thus, a large amount of research is now being undergone, aiming at developing the best lipid-based nanocarriers for oral delivery. This task also takes advantage of the fact that most oils and fats used for the development of these nanocarriers derive from dietary lipids, thus facilitating oral permeability and biodegradability. The term lipid-based nanocarrier includes liposomes, self-nano and microemulsifying drug delivery systems, nanoemulsions and nanocapsules. 

Liposomes are spherical vesicles constituted by lipid bilayers and an aqueous inner core. Their basic composition is phospholipids and sterols (such as cholesterol), with the latter ones being used in order to stabilize the liposomal membrane. However, different components can be added to this simple structure, such as surfactants, bile acids, or specific ligands that could help the targeting of the liposomes to intestinal cells (see below). Moreover, due to their composition, liposomes can carry hydrophilic molecules into their inner cavity, whereas hydrophobic drugs can be inserted into the lipid bilayer [33]. Solid lipid NPs are composed by a lipid core (triglycerides, fatty acids or phospholipids) with a monolayer surfactant shell, such as lecithin or bile salt derivatives [34]. Nanoemulsions are dispersions of an oily and an aqueous phase with the addition of an appropriate surfactant, but due to the percentage of surfactant (3–10%) they are thermodynamically unstable. On the contrary, microemulsions are stabilized by surfactants added in higher concentrations (≥20%), thus making them thermodynamically stable [35]. Lipid nanocapsules are constituted by an oily phase and an aqueous one, stabilized by surfactants and a polymeric shell. Due to their nature, lipid nanocapsules can present with different biological properties, which depend on their surface characteristics. In fact, the characteristics of the polymeric shell can determine the ability of the NPs to interact with the intestinal environment, in particular with the mucus and/or the enzymes present in the lumen [36]. 

MNPs can interact with the intestine either because they are used as therapeutic agents or because they are ingested with food, since they can be used as food preservatives or colouring agents (such as TiO_2_) [37,38]. From the medical point of view, the most extensively utilized are Ag or gold (Au) NPs, but data have also been obtained on palladium, titanium, zinc, and copper ones. Due to their chemical properties, their surface can be easily functionalized to conjugate targeting agents and active biomolecules, and multiple drugs can also be loaded on the same MNP. MNPs have been mainly employed as anticancer agents [39] or to counteract infections, either bacterial or viral [40]. Due to their small size MNPs (in particular Ag and Au) can also perform a passive targeting of cancer cells, i.e., reaching them more easily due to the leakiness of the vasculature growing within the tumour mass. Moreover, MNPs, in particular Ag NPs, are extremely reactive and can interact with many cellular components through the induction of ROS, leading to mitochondrial damage and eventually apoptosis. Although this effect could be quite desirable in cancer therapy, it should be definitely avoided in the interaction with normal cells, in this case the enterocytes [5,41]. 

Polymeric NPs can be of synthetic origin but also made of natural substances, such as polysaccharides; in a biological setting these latter ones are obviously preferred, since they do not provoke or produce toxic effects. Among the natural polymers, the most commonly used are polysaccharides including chitosan, hyaluronic acid (HA), alginates, etc.; due to their chemical structure, they present both hydrophilic groups (necessary for the solubility in water) but also residues able to interact with biological membranes, as further discussed below [42] (Figure 3). 

Among the possible therapeutic NPs, there are also the mesoporous silica ones, made of porous solid material with large surface area and tuneable pore diameters in the range of 2–50 nm. These nanocarriers can be easily synthesized, and they also present the ability to load multiple cargos with different sizes, which can be unloaded in a controlled way. In addition, these NPs are “Generally Recognized as Safe” by FDA, which makes them eligible for therapeutics treatments [43,44,45]. 

### 1.3. Nanoparticles—Intestine Interaction

The gastrointestinal tract, as mentioned, represents a harsh environment for drug delivery, since the active component has to survive the low pH but also cross the intestinal barrier, i.e., the mucus layers and the enterocytes. For this reason, NPs can be functionalized, in order to prevent the attack of pH and enzymes or to favour their passage through the intestine in order to have a systemic effect. 

The protection from low pH can use polymers that have been already employed in the drug industry, such as all the different formulations of the Eudragit^®^, which can be dissolved above specific pHs, thus allowing the drug delivery in the various regions of the gastrointestinal tract, i.e., small intestine or colon [46]. Other molecules can be used to create a shell, and among them there is alginate, which can provide resistance to low pH and, if associated with other molecules, also to enzyme digestion. Alginate is a linear anionic polymer derived from brown seaweed consisting of β-D-mannuronic acid (M) and α-L-guluronic acid (G) linked by glycosidic bonds [47]. The monomer composition can affect the general structure of alginate, making it more rigid and with larger pores (thus with a higher release of the drug) or more soft with smaller pores according to a high or low presence of G blocks, respectively [48,49]. Alginate also responds to pH, and researchers have developed specific emulsions able to swell or shrink according to the environmental pH [50]; in particular, the presence of low pH will maintain alginate in a stable hydrogel form, thus protecting the associated drug, whereas neutral pH will cause the hydrogel dissolution and the release of the active compound. 

Since mucus covers the apical part of the enterocytes, the NPs need to attach to it, but also be able to cross the two different layers in order to reach the enterocytes. In order to design NPs able to deliver their load, several mucus characteristics should be kept in mind; mucins, the main mucus component, contain glycosylated section with negative charges that could bind positively charged NPs, trapping them. Moreover, some part of these proteins are hydrophobic, and this strongly reduces the transport of hydrophobic particles, such as PLGA and polystyrene (PS), which are quite used as NPs. Last but not least, mucins create a sieve-like structure, thus the size of the NPs should also be kept to a minimum. 

Substances employed in NPs can interact with the mucus either increasing NPs ability to adhere to it or augmenting their penetration. 

One of the most used molecules belonging to the first group is chitosan, a nontoxic, cationic polysaccharide derived from chitin (naturally obtained from marine organisms) which has been approved by FDA for biomedical applications. It is biocompatible, biodegradable, and non-toxic; in addition the presence in its sequence of positively-charged N-acetyl glucosamine units favours the binding to the mucus [42]. Chitosan is also rich in hydroxyl, amino, and carboxyl groups, which allow a series of chemical modifications that can increase, for example, its water solubility or its stability [51]. In addition to mucus adhesion, chitosan can induce the opening of the tight junctions, as demonstrated by alteration in the trans-epithelial resistance and by electron microscopy [52,53,54,55]; these opening has been demonstrated to be reversible, at least in in vitro experiments on Caco2 cells, and associated with a redistribution of claudin-4, an essential component of tight junctions [56]. This effect is mediated by a direct interaction between positively charged chitosan and negatively charged integrin aVβ3, fact which causes a conformational change of this latter proteins that aggregate along cell boundaries, reorganization of F-actin and a downregulation of claudin-4 [57]. 

Another molecule widely used in NPs that is able to increase mucus binding is HA, a natural linear glycosaminoglycan, biocompatible, and biodegradable through the action of the host enzymes. Its ability to bind to biological substrates is mediated by the presence of abundant COOH groups [58] and also by the molecular weight (*M*_W_), with a higher efficiency of the adhesion being observed in presence of a lower *M*_W_ [59]. 

The most used substance which helps the passage of NPs through the mucus is polyethylene glycol (PEG) [60]; the addition of this component can change, even in an important manner, the ability of the NPs to cross the mucus layer; Xu et al. observed that, in the case of PLGA NPs, a percentage of at least 5% is necessary to reduce the interaction with mucus, and higher PEG concentrations improved the passage [60]. This could be explained by a “shielding effect” by the PEG molecules, which prevented the interactions between mucin proteins and the NPs core. These data were obtained using a 5 kDa PEG molecule and the authors used as in vivo model mouse vaginal mucosa; the situation could be totally different in the intestine, and also the PEG size could influence the mucus penetration, as demonstrated by Inchaurraga et al., who analysed the effect of different *M*_W_ PEGs on the ability of NPs to reach the enterocytes [61]. Interestingly, better results were obtained using PEG 2000 or 6000, whereas the 10,000 molecule showed a worse performance. Other polymers have been developed, such as poly-N-2-hydroxypropyl methacrylamide (HPMA), a water-soluble polymer with excellent mucus-permeating properties similar to PEG. This compound can dissociate from chitosan NPs during the passage in the mucus, as demonstrated by Liu M et al., but it can also cause an opening of the tight junctions [62]. 

To increase the passage through the intestinal barrier there are, in theory, two possibilities, i.e., cause a loosening of the tight junction or increase the uptake of the NPs by the M cells or the enterocytes. Several compounds can actually interfere with the proteins that are forming the junctions sealing off the intestinal content, such as occludins, claudins, and integrins. Natural food compounds can have an effect on the permeability of in vitro systems, as reviewed by Kosińska et al., and, more recently, demonstrated by Haasbroek et al. that focused their attention on aloe extracts that were able to decrease trans-epithelial resistance in a trans-well Caco_2_ cell model and increase the passage of 4 kDa dextran [63,64]. Chen et al. developed an hydrogel able to adhere to the mucosa and, at the same time, to chelate calcium ions, which are essential for the maintenance of the junctions [65]. They tested these particles, carrying HbS antigen, in mice and were able to demonstrate a higher intestinal immune response compared to the usual vaccination route. Last, but not least, it should be remembered that bacteria causing gastrointestinal disorders are able to secrete toxins acting on the integrity of the tight junctions, causing a damage or a rearrangement of their protein components. Although this kind of intervention could cause the passage of a large quantity of the cargo drug through the barrier, it could be hazardous; for this reason synthetic peptides mimicking the effect of these toxins have been produced. In particular, AT-1002 is a hexamer peptide (FCIGRL) derived from zonula occludens toxin (ZOT) produced by Vibrio cholera [66]. This toxin is able to bind to a receptor present on the apical portion of the enterocytes, activate protein kinase C and cause a transitory disassemble of the tight junctions. It can be added to other NPs components, such as chitosan, as reported in a delivery system for insulin [67] that was able to obtain a good glycaemic control in diabetic rats. It must be kept in mind, however, that increasing the permeability of the junctions could also allow the passage of intestinal antigens; further studies on this aspect must be performed in vivo, although a paper by Sonaje et al. failed to detect an increased passage of LPS following chitosan NPs administration [68].

The passage through cells, either M or enterocytes, could be increased by adding to the NPs peptides that are able to interact with specific receptors present of the apical part of the cells. As regards M cells, studies in mice demonstrated that various types of lectins added to NPs can increase the uptake due to their ability to interact with cellular α-L-fucose moieties [69,70]; unfortunately, these specific moieties are not present on human cells, thus NPs should be functionalized with other peptides. Among them, the Gly-Arg-Gly-Asp-Ser (GRGDS) pentapeptide could be a good candidate, since it binds B1 integrins, present also on human cells. Up to now, it has only been evaluated in a human cell model (Caco2 + Raji), and demonstrated able to increase the passage through Raji cells [71]. Last but not least the route through which NPs can reach other organs can be important in order to avoid the hepatic first pass; for this reason, NPs can be designed to use the lymphatic system, and this means that intestinal absorption should occur through M cells.

As regards enterocytes, several receptors have been described on the apical surface, and known enterocyte-targeting ligands include lectins, transferrin, vitamins, oligopeptides, and monoclonal antibody fragments as summarized in Table 1. The cell entry could also occur through the action of some specific peptides, identified as cell-penetrating peptides (CPPs), that are able to allow the attachment and penetration of the NPs (such as Trans-Activator of Transcription (TAT)), or through a classical receptor-mediated endocytotic process (Table 1 and Figure 3 and Figure 4) [72]. 

Various bacteria-derived peptides can also be used, since they are recognized by TLR4, but these peptides carry the risk of activating the intestinal immune system. Li et al. recently evaluated the possibility of employing a non-toxic form of *Pseudomonas aeruginosa* exotoxin A associated with alginate/chitosan particles; the presence of the exotoxin favoured the transcitosis, but the in vivo administration of these NPs to rats showed that they co-localized with CD11c+ cells, which have an important role in intestinal immune response [86]. In all cases, the NPs and their cargos should be vehiculated to the basolateral side of the cells, thus requiring transcytosis (Figure 2b and Figure 3). This step should not be regarded as trivial, since there is the risk that the fusion of endocytotic vesicles with lysosomes damages the NPs, both in its structure or inactivating the carried drug. For this reason, some researchers developed NPs associated with charge-convertible peptides [78,87]. The presence of these components allows the NPs to survive the acidic pH of late endosomes, since they can act as “sponge” for H+ ions. Last, but not least, the NPs have to cross the basolateral membranes of enterocytes and be released into the circulation; interestingly, Xi et al. observed that the addition of the charge-convertible peptides increased the interaction of the NPs with the proton-coupled oligopeptide transporter present in the basolateral membrane, thus boosting the exocytosis [78]. The interactions between NPs and the intestine could be subdivided in three main categories, i.e., the use of NPs to deliver systemic drugs, which implies the passage through the intestinal barrier and reaching the blood or lymphatic flow, NPs as carriers for drugs that should act on the intestinal mucosa or the “involuntary” interaction due to NPs used as food additives. In this review, we are going to discuss some examples in each category, pointing out advantages and pitfalls. 

## 2. Nanoparticles for Systemic Drug Delivery

The possibility to deliver drugs through the intestinal route rather than using other more invasive ways has been quite captivating for various pharma products, in particular anti-cancer drugs or vaccines. However, due to the large number of the employed molecules and the great differences among the NPs, we decided to focus on a single molecule tackling another disorder, i.e., insulin. Due to the high social impact of diabetes and the need to administer the drug few times during the day, several groups throughout the world have been involved in the development of NPs able to provide the oral delivery of recombinant insulin. 

The most used cores for NPs are polymers, either natural or synthetic ones; among the natural polymers there is chitosan, either alone or in combination with alginate; these NPs have some characteristics that make them suitable for insulin delivery, such as biodegradability, nontoxicity, muco-adhesiveness, and low immunogenicity, as previously described (see Table 2). Other employed natural polymers are HA, albumin, starch (amylose), zein, and lignin, as reported in Table 2.

These natural components were used alone or in combination, in order to exploit the different characteristics of the various components, such as the ability to bind to the mucus, recognize specific cell receptors etc. As regards synthetic polymers, most researchers employed PLGA, due to its characteristics such as biocompatibility, biodegradability, and its common use in in the drug industry, being FDA approved [143]. Its use is also supported by the fact that, when it is broken down, it generates glycolic acid and lactic acid, which are naturally metabolized by the body. Other synthetic polymers that have been used in insulin delivery are polymath-acrylic acid (PMAA), polyacrylic acid (PAA), and polycaprolactone (PCL) (see Table 2). Apart from polymers, also liposomes have been employed for insulin delivery, using liposomes containing bile salts, such as sodium glycocholate (SGC), sodium taurocholate (STC), or sodium deoxycholate (SDC). All the described polymers and liposomes were able to generate NPs with a high rate of incorporation of insulin and to protect the drug from the degradation that could occur in an acidic environment; however, to this basic structure of the NPs, other molecules have been added to improve the adhesion to the mucus/enterocytes and the passage through the epithelial layer. 

The control of insulin release has to be tightly regulated, in order to prevent hyper- or hypo-glycaemic episodes. This concept was extremely important, throughout the years, in the design of the different injectable insulin formulations; for this reason, several research groups designed NPs containing a “glucose sensor”, i.e., a chemical compound able to react to the different glucose level present in the blood. The most commonly used systems are based on glucose oxidase or phenylboronic acid; glucose oxidase catalyses the oxidation and hydrolysis of β-D-glucose into gluconic acid and hydrogen peroxide and, in turn, the production of the gluconic acid lowers the pH within the NPs [144]. The change in pH alters the structure of the NPs, favouring the release of the encapsulated drug; this system not only controls insulin release, but it can also provide a faster release if compared to the same NPs lacking the glucose oxidase and a better glycaemic control in a rat diabetic model, as shown by Chai et al. [111]. The glucose sensing by phenylboronic acid (PBA) can be mediated by two different mechanisms: in the first one, it occurs through a contraction/expansion transition in which glucose binds to PBA altering the balance between its two forms, the hydrophilic and the hydrophobic one and, as a result, the water density in the NPs increases causing the release of insulin. In the second mechanism, called competitive, glucose displaces the drug that was bound to PBA, thus causing its release [145].

Last, but not least, these NPs must be able to provide an in vivo response; although this demonstration has been provided in animal models, as reported in Table 2, trials in humans are still under way (as discussed below), thus, an accurate evaluation of these data will be necessary before these NPs can be moved to clinic.

## 3. Nanoparticles with Intestinal Targets

### 3.1. Inflammatory Bowel Diseases

IBD, which include both Crohn’s disease (CD) and ulcerative colitis (UC), are chronic inflammatory disorders characterized by mucosal immune system dysregulation, which has an impact in the small intestine and colon. In recent decades, the necessity to conceive a novel therapeutic approach to IBD treatment has led to the increased interest in nanobased drug delivery systems [146,147,148]. This is due to the many side effects caused by the commonly used drugs to treat chronic inflammatory disorders, such as IBD. In particular, 5-aminosalicylates (5-ASA), antibiotics, and corticosteroids can cause, in the long-term, several side effects, including bone damage, such as the steroid-induced necrosis of the femoral head. Genome-wide association studies have demonstrated that genetic background is only one of the factors involved in the pathogenesis of the disease together with the environmental ones. The recent advances in understanding the pathways involved in the development of IBD have allowed to provide some more therapies, but since the exact cause is not completely understood, there is currently no cure tackling the primum movens of the disease. However, even classical drugs used to treat IBD could take advantage of new-targeted delivery systems that give the possibility to load drugs, natural compounds, antibodies, and other biological compounds inside functionalized NPs able to reach the colon. On the other hand, as described by Hartwig et al. [149], we must consider that the research on NPs was usually performed considering colonic drug delivery in healthy individuals and not in a pathological condition, so the data should be regarded with caution in IBD patients. In fact, these individuals have important changes in colon microbiota composition and pH mean values, in addition to the diarrhoea that may affect the gastrointestinal transit time. 

### 3.2. Nanoparticles Loading Drugs

Drugs available to treat IBD, such as Budesonide or Prednisolone, could have better efficacy and less side effects if properly conveyed, since a targeted delivery could, in theory, allow to reduce the total amount of drug administered to the patient. Naeem et al. designed a system in which budesonide loading-PLGA NPs were covered by Eudragit^®^s100, thus generating microparticles (NPinMP). Their findings showed that the orally delivered NPinMP in a mouse model of DSS-induced colitis was able to reduce the number of macrophages and neutrophils assessed by immunofluorescence imaging, reduce TNFα serum levels, and cause a restoration of normal colon length. This treatment was superior to the use of NPs alone, which failed to significantly mitigate inflammation; these data can be explained by the better protection through the gastrointestinal tract provided by the double coating of the drug, which allowed a higher quantity of budesonide to be released in the colon [150]. Zhou et al. created a negatively charged Prednisolone-loading nanogel with a high affinity for the damaged colon tissue due to the positive charges located at the inflamed intestinal site. The persistence of this NPs administered by enema in the large intestine was able to provide, through the gradual release, a reduction in inflammatory parameters in a TNBS-induced colitis rat model [151]. Patients with IBD (in particular UC) can also combine the oral treatment with the enema one to achieve better effectiveness. Date et al., tested both nano-suspensions (NS) and micro-suspensions (MS) of budesonide, embedded in an inert mucus substance (Pluronic f127) for the in vivo enema treatment. The particles were tested in TNBS-induced UC mouse model, demonstrating that both formulations were able to restore the colon length and the weight loss. However, the NS showed a better efficacy in decreasing the inflammatory state of the colon, significantly lowering the number of colon-infiltrating monocytes and the levels of pro-inflammatory cytokines within the tissue [152]. In another study, the authors used different drugs (budesonide, vancomycin, and GM-CSF) loaded in NPs composed of human serum albumin covered with heparin; this second coating was chosen since, in theory, it should be able to selectively bind to the inflamed colon area thanks to the negative charges of its glycosaminoglycan molecule. The formulation was delivered by enema in DSS- induced colitis and showed that the NPs can be efficiently loaded with different drugs at the same time. The authors also observed that smaller particles were better retained in a healthy colon, whereas larger particles preferred the binding in the inflamed area, obtaining a reduction in the inflammatory parameters [153]. Lee et al. used Dexamethasone (Dexa) to create spherical polymeric nano-constructs, composed by PLGA and Dexa core and then covered with PEG, for the treatment of IBD. These NPs were injected performing an intravenous infusion to a mouse model of UC. The near infrared imaging results demonstrated the powerful anti-inflammatory action together with the rapid intracellular release of the NPs [154]. Although i.v. administration could be extremely effective, its use in everyday treatment of IBD patients results very difficult, and this kind of approach should be reserved for biologicals. 

Ceria NPs (Ce NPs) are defined as nanozymes since they behave as enzymes with the ability of scavenging multiple ROS types, thus providing anti-redox and anti-inflammatory activity [155]. These nanozymes can exist in both reduced (Ce^3+^) and oxidized (Ce^4+^) state, mimicking, respectively, catalase and superoxide dismutase enzyme activity. In the work of Zhao et al., PEG-loaded Ce NPs were administered in a mouse model of UC, and showed an important reduction in colonic inflammation, as demonstrated by histology and cytokine analysis [155]. In this regard, in a report by Asgharzade et al., Ce NPs were used to deliver Sulfasalazine [156]; Sulfasalazine is the drug resulting from the combination of a sulphonamide and salicylic acid, that are released after the ingestion. Its main mechanism of action includes intrinsic anti-inflammatory and anti-redox activities, and the important reduction in iNOS levels. 

In a preclinical mouse model of DSS-induced colitis, these particles improved the disease activity index, as well as the histopathological score, and upregulated antioxidant molecules, such as glutathione [156]. In another work performed by Ahmada et al., Sulfasalazine was encapsulated in gelatin NPs and was then coated with Eudragit^®^s100. The nanodrug was then orally delivered to mice affected by UC and tested in a cellular model of Caco2 treated with DSS. The major protective effect was observed in 5-ASA NPs compared to the free drug, with an improvement at the histological level, increase in colon length, and decrease in serum inflammatory markers [157]. A model of intestinal organoid has been proposed for the study of IBD, conveying PLGA NPs covered with alginate or chitosan and loaded with 5-ASA. In particular, the alginate and chitosan coating negatively or positively charged the NPs. As expected, chitosan-covered NPs were preferentially transported through the epithelium to the intestinal organoid lumen [158]. 

Interestingly, the possible application of drugs not currently used to treat IBD, delivered in form of NPs, could open other possibilities in the treatment of the disease. Some of these drugs have a role in the modulation of the inflammatory state, such as Isoniazid (INH), an anti-tuberculosis drug known to have important anti-inflammatory actions and a structure similar to COX II inhibitors. The agent was entrapped into an enteric polymer Eudragit^®^s100, that was degraded at colon pH 7. In the DSS-induced colitis in mice, the comparison of the effect of the free drug and the drug-loaded NPs, assessed by H&E staining, revealed the restorative effect of the NPs-loaded with INH as compared to the free drug. In addition, the authors also demonstrated a possible synergistic effect of the nanodrug in combination with 5-ASA [159]. Another example is Raloxifene, an anti-cancer drug that modulates the estrogen receptor; Greish et al. proved its inhibitory effect on the pathway of NF-kB, a central player able to regulate the production of inflammatory cytokines in IBD. Their report compared the use of the free drug with the drug loaded on PS co-maleic acid micelles, testing them on in vitro and in vivo IBD models. Both formulations were able to induce a protective effect downregulating the NF-kB-dependent signalling pathway, even though the NPs-associated drugs had a major inhibiting power, particularly in lowering the production of IL-6 and TNFα [160]. Cai et al. studied a pH responsive system based on the administration of Tacrolimus, a calcineurin inhibitor that regulates the expression IL-2 and T cells signalling. The drug was loaded onto chitosan NPs functionalized with tripolyphosphate (TPP), a polyanion linked together by crosslinking, HA (with high affinity for the CD44 receptors), and Eudragit^®^s100 as enteric coating material. In vivo, their experiments suggested that the orally delivered NPs were able to restore colon length, reduce histological damage and prevent the development of the inflammatory cascade typical of IBD [161]. Antibiotics can also have a role in IBD by changing the microbiota composition, which could be altered in this disease; moreover, a subsequent targeted supplementation could help to restore a normal microbiota diversity. One example is Rifaximin a non-systemic antibiotic with antimicrobial capacity, which was loaded on tamarind gum NPs. These NPs were able to resist the degradation of the upper intestinal tract and showed a mucus adhesive capacity in the colon, allowing a prolonged release of the loaded drug. This experiment was carried out on Wistar rats with TNBS-induced colitis showing that these NPs were able to improve the colon length and decrease the serum levels of inflammatory cytokines as compared to the not-treated rats [162]. 

### 3.3. Nanoparticles Loading Biologics

Today, biological drugs used to treat IBD patients include anti-TNFα antibodies, as well as anti-interleukins and more recently anti-integrins antibodies. TNFα is known to be the key molecule in the development of the uncontrolled inflammatory response in IBD and, for this reason, several works have tried to identify new ways of vehiculating anti-TNFα antibodies [163]. Currently, biologics have some limitations, such as the i.v. administration route, that is useful to obtain a systemic effect, but in the meanwhile favours adverse reactions; moreover, the treatment can be administered only under medical supervision with a decreased patient compliance. The COLOPULSE coating is a pH-responsive polymer consisting of an integrated sodium croscarmellose matrix and is released immediately after the digestive passage from the jejunum to the ileum, where the pH is around 7. The formulation carried an anti-TNFα antibody (Infliximab) and was tested in the gastrointestinal simulation system; this study proved that the COLOPULSE system was able to release the drug at the selected pH, and the authors suggest its possible therapeutic use for ileo-colonic IBD patients [164]. In a recent study, Wang et al. orally administered Infliximab in tannic acid 1,2-distearoyl-snglycero-3-phosphoethanolamine-N-[methoxy(PEG)-2000 NPs in a mouse model of UC. The treatment was performed immediately after the DSS-mediated damage, and ameliorated the disease activity index, also decreasing the production of inflammatory cytokines. These results suggested that loading the antibodies on specific NPs could allow a better delivery of biologicals to the inflamed colonic area [165].

An alternative approach in the treatment of IBD is to decrease the TNFα expression using antisense oligos or siRNAs, which can be easily orally delivered, thus maximizing the beneficial effects for the patient. Knipe et al. proposed a TNFα siRNA loaded in polycationic 2-(diethylamino) ethyl methacrylate-based nanogels and then encapsulated within a poly-(methacrylic acid-coN-vinyl-2-pyrrolidone) hydrogel, which can be degraded by enteric enzymes. It was demonstrated that the loading efficiency of siRNA was greater than 90%, and that the hydrogel was enzymatically degraded, thus releasing the siRNA, in the presence of simulated gastric fluid. Moreover, the delivery of the designed compound to RAW 264.7 macrophages demonstrated its efficiency in knocking down the TNFα levels [166]. It must be noted, however, that the efficacy of the “digested” NPs was only tested on a macrophage cell line, not on intestinal tissue where the release of the antibody should occur. In a genome wide association study comparing healthy individuals and patients affected by UC, different miRNAs were found to be upregulated in presence of intestinal inflammation. miR-31 was detected as involved in the early stages of the inflammatory response, with a direct effect on Wnt and Hippo pathways’ target proteins. Tian et al. observed that administering via enema oxidized konjac glucomannan-peptosome microspheres loaded with miR-31 to DSS- or TNBS-treated mice was able to improve the intestinal inflammatory response [167]. miR-223 was found to be a key control check for NLRP3 inflammasome and also to be able to support intestinal homeostasis. Neudecker et al., reported an improvement in weight loss and colon length, after the i.v. administration of 1,2-dioleoyl-sn-glycero3-phosphocholine NPs with squalene oil, polysorbate 20, and an antioxidant, loaded with mmu-miR-223 to mice with DSS-induced colitis [168]. Li et al. used exosomes derived from human mesenchymal cells, employing them to convey miR-181a in a model of UC induced in mice and in human colonic epithelial cells. According to miR-181a protective function, inducing its expression resulted in an improved colon pathology with a decreased in TNFα expression [169]. Regarding CeD, siRNAs are currently being studied for the creation of drugs targeting the pathophysiological pathways. Attarwala et al. [170], showed that the Transglutaminase-2 and IL-15 siRNAs charged on Au NPs were able to induce silencing of both targets in a cellular model of Caco2. The administration of nanoparticle-in-microsphere oral system (NiMOS) has been used for conveying or transfecting nucleic acids or siRNAs in specific regions of interest. Administration of IL-15 and IL-15 TG2-NiMOS showed an improvement in inflammatory condition in mice Poly (I:C)-induced enteropathy, suggesting another possible approach to treat the disease [171]. Dong et al., used *Staphylococcus* nuclease (SNase) to understand if its administration was able to induce neutrophil extracellular traps degradation and could then improve DSS- induced UC in a mouse model. SNase was trapped along with Calcium alginate via crosslinking reaction and was orally administered as a potential therapy in the in vivo model. The reduction in mRNA levels of inflammatory cytokines in the colon, such as TNFα, IL-6 and IL-1β was shown to be the protective effect of SNase-alginate, as well as the increase in serum levels of anti-inflammatory cytokines IL-10 and IL-27 [172].

### 3.4. Colon Cancer

Unfortunately, the toxic effects of chemotherapeutic drugs are not only restricted to the tumour cells, but also act at systemic level. For this reason, NPs functionalization aims to develop a more targeted drug delivery system with the drug release only in the presence of tumour tissue, thus reducing the side effects related to the unspecific targeting of healthy cells. 

Colon cancer, particularly colorectal cancer (CRC), is the third most common diagnosis of cancer and the second most common diagnosis for both sexes. 5-Fluorouracil (5-FU) is an analogue of pyrimidine that acts as a timidylate synthase inhibitor, as in the major used drug in CRC treatment. The delivery of a mesoporous silica NPs loaded with 5-FU and entrapped in a galactosylated-chitosan polymer has been proposed to treat human colon cancer and tested in vitro on SW620 cells. The modified chitosan was able to increase its-potential, the loading capacity of the NPs, and facilitated galactose receptors uptake by SW620 cells, as demonstrated by fluorescence microscopy and flow-cytometry analysis [77]. Metastatic cancer requires systemic chemotherapy that is challenging to sustain for more than 50% of patients. Oxaliplatin, as a chemotherapy agent, blocks cell replication and leads to cancer cells death. This, together with 5-FU, has been shown to increase the survival rate in CRC patients. Iron Oxide NPs (Fe_3_O_4_) are known to undergo magnetization and have a high relaxivity (ability of Fe_3_O_4_ to enhance the intensity of magnetic resonance images), characteristics useful for hyperthermia applications, in which the heat stimulus is driven by alternating magnetic field (AMF) and leads to cell death. Fe_3_O_4_ and Oxaliplatin loaded on liposome NPs (L-NIR-Fe_3_O_4_/OX) were tested on CC-531 cell line and on mice with orthotopic tumours. In vitro, within one hour, these NPs were uptaken by cells and caused increased cell death. Moreover, the in vivo use of these NPs followed by the external AMF stimulus improved survival in colorectal liver metastatic tumour in rats [173]. The purpose of Saber et al. was to evaluate a possible combination therapy via nanocubosomes formed by emulsification of glyceryl monooleate (GMO) and Pluronic-F127, in which cisplatin (CPT) and metformin were added. Subsequently, they delivered the NPs to human CRC cells HCT-116 for efficacy and toxicity studies and observed a stronger anticancer activity of the drug-loaded NPs when compared to the free CPT. The NPs formulation was able to affect different metabolic pathways decreasing ATP and glucose levels and inhibiting mTOR/Akt pathways, which led to apoptotic cell death. Finally, they demonstrated that the combined NPs had a better efficacy due to the combination of the two drugs loaded into them and their synergic mechanism of action [174]. The chemical conjugation of a drug with squalene is possible thanks to its conformation that allows the spontaneous self-assembly of the NPs in water. CPT was loaded on the hydrophilic NPs and then ex vivo efficacy was tested on human colon cells and on ApcMin/+ mice treated with azoxymethane/dextran sulphate sodium to induce intestinal tumorigenesis. The NPs showed to be 10 times more efficient in inducing cancer cell death than the free drug, with an increase in the intracellular accumulation. Similarly, decreased tumour formation and no apparent systemic NPs toxicity was found in the in vivo model [175]. Methotrexate (MTX) is an analogue of folic acid as it has an amino group instead of a hydroxyl group in the 4′ position. This conformation allows it to undergo the same folate transport system, binding to the respective receptors and inhibiting dihydrofolate reductase. For these reasons, in a study by Álvarez-González et al., Au NPs were used as carriers, replacing the reducing agent with the carboxyl group of MTX. The NPs loading MTX were tested on colon cancer cells HTC-116, showing a good dose-response effect, enhancing cancer cells apoptosis, and inhibiting the folic acid cycle usually upregulated in tumours. Moreover, AuNPs conjugated to MTX were shown to have higher cytotoxic activity compared to the free drug [79]. Colon adenocarcinoma corresponds to 2/3 of all colon tumours. Depending on the patient’s condition, current therapeutic recommendations include chemotherapy treatments, resection of the tumour or immunomodulatory agents’ combinations. Scheetz et al. proposed a co-delivery system, including a chemotherapy drug docetaxel conjugated to a TLR9 agonist CpG oligonucleotide, loaded together on synthetic HDLs. The use of HDLs allowed cells expressing the Scarb-1 (SR-BI) receptor to take up the particles and to reach a high cell accumulation. Synthetic HDLs have an average diameter of 8 to 14 nm and this allows them to have an efficient trafficking at systemic level compared to synthetic NPs. The formulation was tested on MC38 cells derived from mice with colon adenocarcinoma, demonstrating a good tolerability and a strong cytotoxic effect with the preservation of normal cytokines serum levels [176]. Among the latest innovative therapies for the treatment of cancers, siRNA-based RNA interference and gene silencing systems have emerged. Xiao et al. targeted the transmembrane protein CD98 type II, known to be over-expressed in the development of colon cancer, by creating Fab’siCD98/Camptotheicin NPs. The NPs used to load the compound were a hydrogel composed of alginate and chitosan. The authors tested the hydrogel in vitro in human colon cancer cells and demonstrated that the drug was efficiently released and had a major therapeutic efficacy compared to the non-functionalized version [75]. A paper by Li et al. focused on lncRNA NEAT1 that is known to be upregulated in colon cancer and promote its progression, as well as that of other tumours. It has been shown that the expression of NEAT1 is regulated in both cancer and para-cancerous states and that targeting NEAT1 causes a strong inhibition of tumour growth, accelerating the apoptotic process of CRC. The authors tested chitosan-based NPs loaded with lncRNA NEAT1 siRNA on human HCT-116, Lovo and SW480 colon cancer cells, as well as a normal NCM460 colonic cell line, observing an important growth inhibition in all the colon cancer cells [177]. These latter approaches suggest that the use of NPs could be extremely helpful for the delivery of “molecular therapies”, preventing their degradation and targeting them to the cells that have to be destroyed; it remains to be determined, however, whether this approach will be feasible only for the i.v. route or also for a more localized treatment. 

## 4. Clinical Trials Status

Since NPs as drug delivery system offer a promising approach to obtain targeted and specific effects on human diseases, there have been several NPs approved by the FDA and the EMA starting from 1989. However, the incomplete international regulatory guidelines and the lack of standardized physiochemical characterization protocols still represent critical challenges in clinical studies. Although a large proportion of approved NPs has been realized for i.v. administration, great efforts in the research field have been carried out to extend nanomedicines to other administration routes, particularly the oral one (see Figure 1). According to a recent review by Halwani [32], until today the marketed nanomedicines can be classified as polymer-based, lipid-based, inorganic, dendrimer, and protein-based NPs; a summary of the approved NPs, their composition and administration route is reported in Appendix A. Among these NPs, the majority of them is intended for cancer therapy, antimicrobial agents, autoimmune conditions, and gene therapies.

As reported in www.clinicaltrials.gov (accessed on 4 April 2022) there are several ongoing clinical trials aiming to expand the original clinical indication for already approved NPs; Liposomal Doxorubicin is one of the more tested lipid-based NPs counting a total of 593 trials (355 “recruiting” and 238 “active not recruiting”), liposomal Vincristine counts 342 trials (191 “recruiting” and 151 “active not recruiting”) and protein-based NPs as albumin-bound Paclitaxel counts 1310 total trials (833 “recruiting” and 477 “active not recruiting”). ThermoDox is an example of an already existing liposomal-loaded Doxorubicin, approved for the treatment of metastatic ovarian cancer and breast cancer, which was modified in order to obtain a heat stimuli-responsive drug (NCT00617981 completed Clinical Trial) [178].

Different kinds of new NP-based molecules, in particular insulin-based NPs, are currently reported in www.clinicaltrials.gov (accessed on 4 April 2022) (Table 3). 

However, due to the important intra and inter-subject variations in oral insulin bioavailability that may be due to sensitivity to intestinal proteases and decreased penetration into intestinal epithelium, the best insulin-NPs functionalization has not been found yet. At the present time, 19 completed clinical trials employed oral formulations of insulin loaded-NPs; unfortunately, 15 studies did not report the final results, thus preventing an evaluation of the performance of the drug. ORMD-0801 is an example of human insulin NPs that underwent several clinical trials, among which the phase II NCT02496000 study. During this trial, patients with Type 2 Diabetes were treated with insulin-NPs, observing an important reduction in insulin without hypoglycaemic side effects [179,180].

## 5. Food Additives

Along with their therapeutic use as drug carriers, NPs are also used as additives in the food industry, mainly as colouring, flavouring, or texture-improving agents, but also for their antimicrobial properties. A small fraction can also be used in food packaging. The two main categories used are metallic and silica-based NPs and the characteristics usually evaluated to determine their safety regard their unintended effects on the human body for their potential accumulation in tissues, their distribution, and excretion routes. The main concerns for MNPs regard the potential genotoxicity and accumulation, the release and accumulation of metallic ions once in the body, as well as NPs aggregation with food matrix, or the aggregation with digestion enzymes and bile salts, that could impair a proper digestion and assimilation of nutrients. NPs could also have an impact on the inflammatory status of subjects with an already altered intestinal homeostasis, such as patients with IBD or CeD. The potential impact on the microbiota and the oxidative stress potential are nevertheless relevant [38,181,182,183]. 

In the US, food additive use is regulated under the FDA, that applies a case-by-case policy, with, for example, up to 1% of E171 allowed on the final weight of the product, or <5 ppm for E172 [184]. In Europe, food NPs were allowed *ad quantum satis*, but the guidelines for determining the safety of NPs as food additives have been revised in 2021, along with EFSA(European Food Safety Authority) position statements about some specific NPs [185]. In particular, E171 (TiO_2_) presence in food has been considered unsafe, for the growing evidences of adverse effects in in vitro and in vivo studies, leading many European nations to ban E171 as food additive in early 2022, even if it can still be an ingredient for drugs and supplements [37]. ZnO NPs were instead regarded as safe to use in food packaging, since their release in the contained food does not happen as NPs, but rather as Zn ions [186]. The migration of these ions complies with the current migration limit, but it needs to be considered that the daily limit of assumption of 25 mg/person could be exceeded, in combination with dietary exposure. Similarly, Ag NPs are considered safe when used in food packaging, since the diffusion of Ag ions from packaging is estimated to be very low and below the limit of acceptable daily intake of 0.9 ug, whereas its use as food additive (E174) is allowed even if still controversial [187,188]. In fact, data are still considered incomplete, although there is an increasing amount of studies about this element. Similarly, Au NPs (E175) are allowed, since there are not enough data on its adsorption, distribution, metabolism, and excretion that could determine its unsafety, whereas for E173 (aluminium based), normally used in confectionary, there is a recommendation not to exceed a weekly intake of 1 mg/kg body weight, as it can accumulate in the organism and aluminium persistence has been indirectly linked to neurological conditions [189,190]. E551 is instead SiO_2_ based, but its composition is various since it is known that it can include a fraction of NPs, but not all the other components are specified [191]. In EFSA latest update, it was recommended to further characterize its composition and to lower the current limits for toxic elements in E551, mainly to avoid the presence of contaminants that are regarded as harmful (arsenic, lead, mercury, and cadmium). Table 4 summarizes the main NPs-containing food additives and their current assessment according to EFSA. 

Of note, the main reason for many NPs not having limitations, is because there are not enough studies about their behaviour once in the human body, including the potential side effects. Figure 5 highlights the main potential mechanisms of toxicity of MNPs in the intestine.

Here, below, we will deepen some relevant aspects about the NPs more studied at the moment, namely Ag NPs and the recently considered unsafe TiO_2_, mainly focusing on the most recent studies.

### 5.1. E171—Titanium Dioxide

TiO_2_, the food colouring agent E171, is usually composed by less than 50% of particles with <100 nm dimensions. Being the NPs fraction a consistent part of the additive and the one that raises more concerns, works using only the TiO_2_ NPs (usually anatase 99%) were considered reliable for the analysis of E171 safety [37]. Being used as a food whitening agent, it is estimated that children exposure is even more relevant than that of adults, due to the major consumption of sweets and processed products. The main reasons that led EFSA to no longer advise E171 as safe are mainly due to its potential genotoxicity, since it has been demonstrated that it can induce DNA strand break and chromosomal damage [182]. At the same time, many concerns about potential immuno-toxicological and inflammatory effects mainly involving the digestive system, as well as accumulation in tissues, contributed to the decision. Different aspects have been considered, starting with the difficulty to set up a proper model that could replicate the possible changes that E171 undergoes during the digestion, involving also its interaction with the food matrix. In fact, the different pH and ionic strengths encountered in different districts (mouth, stomach, intestine), can cause aggregation or agglomeration, states that could improve or worsen TiO_2_ reactivity. According to Murugadoss et al., smaller TiO_2_ NPs can agglomerate when exposed to acid pH and result in impaired cell permeability in Caco2, an increased oxidative status and the production of inflammatory cytokines in THP-1 cells, whereas a clear increase in DNA damage was observed in mice [193]. A strong aggregation mainly induced by ionic strengths (ions presence) in gastric and duodenal fluids was instead observed by Marucco et al. in their model of simulated human digestive system (SHDS) [194]. They also appreciated a relevant protein corona formation when proteins were added to the fluids, which could partially modulate the significantly increased ROS production in HCT116 colon cells exposed for 24 h to the different obtained NPs aggregates. Cao et al. had previously set up a similar model, with slight differences in fluid compositions, testing the resulting NPs aggregates on a tri-cellular model mimicking the intestinal cell populations [195]. Opposite to what observed by Marucco et al., the authors appreciated a more relevant production of ROS when TiO_2_ NPs aggregated with food matrix, but in a shorter period (6 h) and with higher TiO_2_ concentrations. They also observed a more pronounced cytotoxic effect after 24 h, when TiO_2_ NPs were not aggregated with food matrix. The potential interaction with digestive enzymes was instead investigated by Dudefoi et al., since they included them in their SHDS and observed large agglomerates mainly in duodenal fluids with α-amilases and divalent cations [196]. It was estimated an impairment in amilases function of about 10% in saliva and in the duodenum, which could have a slight impact on the digestion of carbohydrates. Bile salts can also reversibly adsorbe on TiO_2_ NPs, potentially interfering with the digestion of lipids [197]. Since TiO_2_NPs are not degraded with the digestion, Coméra et al. analysed its eventual intestinal uptake after single oral administration in vivo, finding a peak of absorption in the jejunum and the ileum of mice after 4 h, whereas a peak of TiO_2_ was seen in jejunal Peyer’s patches and in peripheral blood after 8 h [198]. Indeed, TiO_2_ can pass through the intestinal barrier both by cellular uptake and by the paracellular route, also reaching different tissues where they could accumulate, such as liver, lungs, kidneys, and reproductive organs. TiO_2_ has in fact been found in the placenta of pregnant women and in the meconium of the newborns [199], whereas studies in mice and rats highlighted neuro-toxicological and neuro-developmental effects in the offspring, as well as a chronic low grade inflammation in their intestine [200,201]. In this regard, Bettini et al. already demonstrated that the chronic exposure to E171 in rats could induce a systemic inflammation and an inflammatory microenvironment in the intestine, with the production of IL-1β, IL-8, and TNFα [202]. This low-grade inflammation could not be a problem in healthy subjects, but could also exacerbate pre-existing conditions, such as CeD or IBD. In fact, several years ago Hummel et al. already identified the presence of TiO_2_ in Peyer’s patches of children with IBD, even if their effects were not clear [203]. More recently, Mancuso et al. observed instead an increase in the inflammatory response in intestinal biopsies of celiac patients when treated with TiO_2_ and gliadin (CeD exogenous antigen), as well as rearrangements in the tight junctions of Caco2 cells exposed to the combination of TiO_2_ + gliadin [6]. This loosening of the tight junctions could indeed impair the intestinal barrier functionality, increasing the permeability and helping the translocation of peptides and bacteria in the sub-epithelial space, favouring the “leaky gut” condition common in IBD [204]. Concerns about this aspect were already raised by Talbot et al. in 2018, since they observed the trapping of E171 in the mucus layer, even if without consequences on intestinal functions in healthy mice [205]. On the other hand, Zhu et al. appreciated a strong dysbiosis-induced intestinal inflammation, associated to the decreased thickness of the mucus layer in a model of obese mice fed with high fat diet and TiO_2_ [206]. Similarly, Cao et al. observed colonic inflammation (mediated in particular by IL-12 and IL-17) and a decrease in short chain fatty acids in mice exposed to TiO_2_, especially when obese [207]. They also demonstrated that the intestinal inflammation was mediated by intestinal dysbiosis, since healthy mice that underwent faecal transplant with the material of TiO_2_-exposed ones, ended up developing colon inflammation. Moreover, mice with metabolic syndrome had increased intestinal inflammation and liver damage, which were worsened by the concomitant exposure to TiO_2_ [208]. Interestingly, probiotics containing *Lactobacillus rhamnosus GG* increased tight junctions and restored the intestinal barrier functionality, with a protective effect similar to what already seen regarding TiO_2_-induced inflammation in rats [208,209]. Finally, all the studies performed on this food additive led EFSA to the decision of no longer advising E171 as safe in food, even if there is not similar advice for drugs and supplements at the moment. 

### 5.2. E174—Silver NPs

About Ag NPs, EFSA distinguished their use as food additive (E174) or as anti-microbial agent in food packaging [187,188]. If their use embedded in polymers does not raise many concerns, there are still points to be addressed for E174. First of all, E174 composition needs to be elucidated, thus characterizing Ag NPs in food is a first important step. For this reason, Corps Ricardo et al. developed a method for the screening of nanomaterials in food, that could help the detection of NPs based additives, and validated it for Ag NPs and its quantification in processed food, in order to improve food safety assessment [210]; in addition De Vos et al. characterized E174 content of Ag NPs from different E174 sources and products [211]. Ag NPs use has increased in many sectors due to their versatile applications, but it has been observed that their presence in the environment can cause toxic effects on wildlife, especially for aquatic animals and seafood [212,213]. Therefore, Ag NPs potential side effects once in the human body cannot be underestimated. Possible problems related to Ag NPs regard their oxidative potential, genotoxicity, cytokines induction and the impact on microbiota [11,214,215,216]. Although it needs to be noted that the concentration of NPs used for cell stimulation in different studies on E174 effects was various, ranging from about 1 to 100 ug/mL, the exposure to this additive was not without consequences.

Similarly to TiO_2_, Mulenos et al. observed low ions release also for Ag NPs under different conditions, leading to the hypothesis that Ag NPs could exert cell damage themselves [217]. In vitro experiments on intestinal cells highlighted increased ROS production and inflammatory response; for example, the oxidative stress response leading to IL-8 secretion in Caco2 cells after 21 h of exposure to Ag NPs [218], or the increased presence of ROS and consequent increase in the Bax/bcl2 ratio seen in cancerous (HCT116) and non-cancerous cells (NCM460) [219]. This effects could be partially modulated by the use of flavonoids, capable of protecting Caco2 from Ag NPs citotoxicity and preventing neutrophil oxidative burst, considering also the increased susceptibility to cytotoxic effects with ongoing inflammation [220,221]. Moreover, dysbiosis in gut microbiota, similar to the one of obese subjects, was observed in mice exposed to Ag NPs [222], whereas in vitro studies showed that microbiota changes due to Ag NPs were more relevant than for TiO_2_ [223], and that the concomitant use of probiotics can reduce their impact [19]. Even more importantly, in ex vivo studies Gokulan et al. tested Ag NPs on the terminal part of the ileum excised from human subjects, obtaining an increased response of different cytokines (such as IL-1β, IL-6, IL-8, and IFNγ) in male and female subjects, even if with high variability [224]. 

Ag NPs can also interact with food components, situation that could be detrimental since this combination can alter the chemical properties of both players. In fact, a barrier impairment was observed in Caco2 cells exposed to the combination of Ag NPs and gliadin, as well as an increased inflammatory response in intestinal biopsies of celiac subjects on a gluten free diet [6]. On the other hand, there are studies testing Ag-based NPs for therapeutic use, also in IBD, and a protective impact was also observed for TiO_2_ in combination with vitamin E, but the possible administration of a NPs-based drug would happen after a validated trial and with a controlled posology, whereas the casual exposure to food additives can be uncontrolled [225,226]. To conclude, similarly to E171, E174 may not be a problem in healthy subjects with proper antioxidant defences and a balanced microbiota, but could be deleterious in subjects with an already altered homeostasis. 

## 6. Summary

The published data reviewed in the present manuscript underline the great opportunities provided by NPs in the therapeutic field, such as the possibility to develop new formulations for the oral delivery of drugs which could be previously administered only by injections. On the other hand, it also discusses the possible side effects of NP-delivery, i.e., effects on the microbiota or intestinal toxicity. 

The development, in the last years, of a series of NPs with different chemical composition (metallic, lipid-based, polymeric) has widen their applications in the medical field, since NPs-bound drugs can be more stable but, above all, can be targeted to specific organs or cells, gaining also the ability to cross specific barriers, such as the intestinal or the blood–brain ones. The possibility to provide a specific targeting is essential in the case of drugs that could otherwise be toxic (such as chemotherapic agents), but it could also be exploited to develop new oral formulations that, being regarded as less invasive, could increase patients’ compliance. In fact, the attention of researchers and pharmaceutical companies has focused on the development of insulin oral delivery, but also on the treatment of IBD. As regards insulin oral treatment, several different NPs containing molecules able to increase the intestinal passage, as well as glucose sensors have been developed, and there are currently ongoing clinical trials on diabetic patients. 

These promising results represent, however, the bright side of the possible interactions between NPs and the intestine; in fact, the introduction of NPs through the oral route could alter the microbiota composition but also the tight junctions or, in the case of MNPs, be able to cause direct damage on the enterocytes. These side effects apply not only to the therapeutic NPs, but also to those present in food either as colouring agents or preservatives. The alteration in the microbiota composition can, in turn, cause an abnormal production of microbiota metabolites, which can affect the general well-being of the individual. On the other hand, the damage of the intestinal mucosa could favour the passage of lumen molecules, triggering an inflammatory/immune response, essential component of disorders such as CeD or IBD. Last, but not least, the possible accumulation in the body of NPs should be also considered in order to evaluate their potential toxicity. 

## 7. Future Perspective and Conclusions

The studies presented in this review recap the main NPs applications in the huge field of nanomedicine: NPs functionalization for systemic drug delivery, the characterization of drugs-nanocarriers to target the intestinal tract and the main consequences of food additives ingestion in daily life. Nanodrug delivery system have a huge potential from the clinical point of view; in fact, some of the cited drug-loaded NPs are currently under clinical trial evaluation by FDA. Although we focus on the specific interactions between NPs and intestine, there are currently numerous studies that are focusing on the use of NPs, in particular to i.v. administration, to pass other barriers normally present in our organism. Obviously, the most interesting one would be the blood brain barrier due to the difficulty to have a targeted delivery to this organ. Thus, a possible and desirable future development for NPs would be to create nanocarriers, which could be orally administered and targeted to specific organs. Due to the huge number of NPs already described or under study, it is quite difficult to identify the possible best candidates for future therapies, also because these NPs need to be tailored to the target organ. However, there are some general considerations, which could provide food for thought for future developments. The creation of NPs able to bind and vehiculate different molecules could represent a definite advantage, both to improve the therapy as well as to reduce the potential toxicity of NPs themselves. In fact, in several diseases it is often necessary to use combined therapies, and the possibility to load them on the same carrier could represent an advantage for the patient. On the other hand, as described in this review, NPs can induce toxicity through the generation of ROS and oxidative stress, thus the insertion of antioxidant molecules in the NPs could reduce the unwanted cellular damage. Another issue that would require further evaluation is the possibility to target oral NPs either to the enterocytes or to M cells; this can determine the fate of the NPs, which would be directed to the portal blood or to the lymphatic system, respectively. The first route can be exploited for the delivery of liver-specific drugs aiming, for example, to treat common liver diseases, such as non-alcoholic fatty liver or non-alcoholic steatohepatitis. Conversely, the use of the lymphatic route will represent an advantage for oral vaccination, with NPs-transported antigens. Lastly, for NPs developed for IBD treatment, apart from the characteristic cited above (combination of different molecules), a more specific targeting to the inflamed tissue and a longer release could represent a plus. In this field, however, further studies are needed, also considering that the classical IBD animal models develop colitis after chemical treatment, thus not recapitulating all the processes involved in the pathogenesis of these disorders. 

On the other hand, our attention should also be pointed at the environmental exposure to NPs, which became quite inevitable in the last years, with effectively poor awareness of the potential adverse effects. Finally, although NPs represent a delivery system that could become extremely useful even for more advanced therapeutic approaches (such as gene therapy) further efforts should be made in the near future to gain a more complete understanding of the therapeutic effects of nanotechnology, keeping in mind at the same time all the related safety issues. 

## Figures and Tables

**Figure 1 ijms-23-04339-f001:**
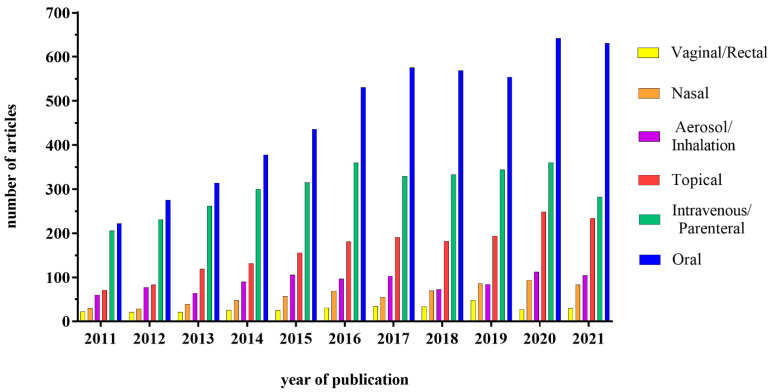
Graphical representation of the number of articles published on the therapeutic use of NPs subdivided according to the administration route. Database used: PubMed; Keywords used in advanced search: “Nanoparticles + “oral”; “intravenous OR parenteral”; “topical”; “vaginal OR rectal”; “aerosol OR inhalation”; “nasal” + drug delivery.

**Figure 2 ijms-23-04339-f002:**
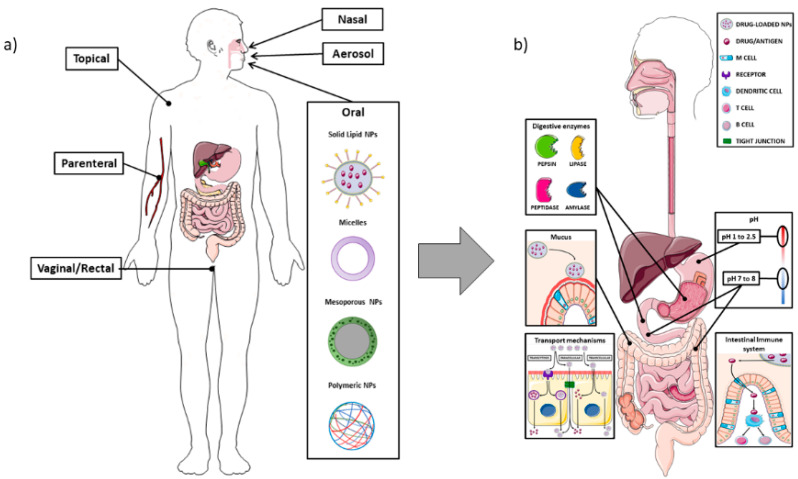
Schematic representation of the different routes for nanoparticle drug delivery, with attention to the oral administration and the interactions with the intestinal barrier. Panel (**a**): Overview of the main administration routes for nano-drug delivery; Panel (**b**): Different enzymes (pepsin, lipase, peptidase, and amylase) located in the gastrointestinal tract can impair nanocarriers stability and their ability to reach the target tissue. The mucus layer also plays an important role in the entrapment of NPs, which may lead to reduced uptake at cellular level. The enterocytes transport mechanisms of NPs can occur through the intestinal cells, either by transcytosis (mediated by endocytic vesicles), or through a direct apical-basolateral passage, or by the paracellular route (passing through the intercellular space). The difference in pH among the stomach, duodenum, and colon represent one of the main challenges in delivering NPs, particularly in order to avoid their premature degradation through the acidic environment. M cells, as part of the GALT (gut-associated lymphoid tissue), can detect antigens from the intestinal lumen and bring them to antigen presenting cells (APC), which, in turn, are able to present them to B or T lymphocytes located at the mucosal level. The image was created with the use of Servier Medical Art modified templates, licensed under a Creative Common Attribution 3.0 Unported License (https://smart.servier.com, accessed on 19 February 2022).

**Figure 3 ijms-23-04339-f003:**
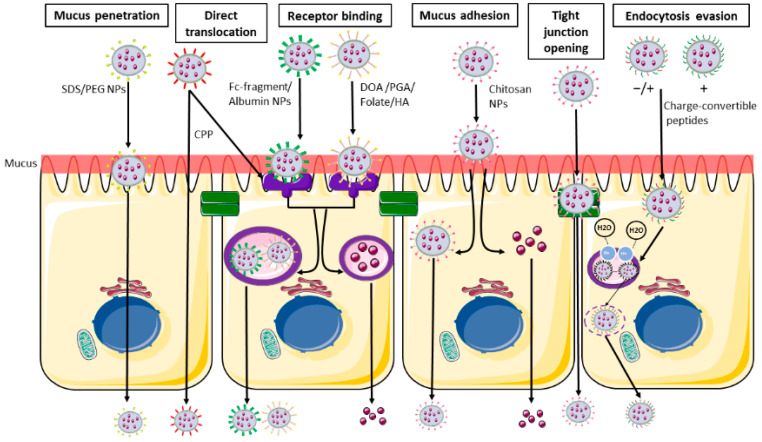
Main nanoparticles functionalization and their intestinal transport. From the left: Schematic representation of mucus penetrating NPs (SDS/PEG), able to penetrate the mucus layer and directly pass through the blood flow. Receptor binding NPs (DOA/PGA/folate/HA/albumin/Fc-fragment) able to bind the cell surface using the ligand-receptor binding and are then internalized in endocytic vesicles and released in the systemic circulation. CPP (cell penetrating peptides) are able to undergo both receptor binding internalization and direct translocation. Muco-adhesive NPs and tight junction opening NPs (chitosan) are able to be retained in the mucus layer, and then undergo transcellular passage or pass through the opened tight junction. Charge-convertible peptides are able to evade the lysosomal degradation using the proton sponge mechanism. SDS: sodium dodecyl sulphate; PEG: polyethylene glycol; CS: chondroitin sulphate; DOA: deoxycholic acid; PGA: poly-glutamic acid; HA: hyaluronic acid. The image was created with the use of Servier Medical Art modified templates, licensed under a Creative Common Attribution 3.0 Unported License (https://smart.servier.com, accessed on 19 February 2022).

**Figure 4 ijms-23-04339-f004:**
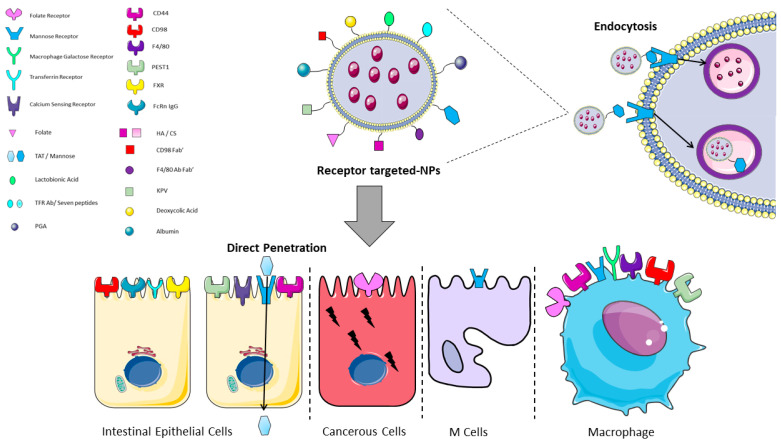
Graphical representation of receptor/ligand NPs interactions in the intestine. Functionalized receptor-binding NPs are able to bind the cell membrane through the binding of the NP (ligand) to the receptor on the cell surface and then to undergo the endocytotic process. Trans-Activator of Transcription (TAT) is the only mentioned ligand that undergoes direct penetration. CS: chondroitin sulphate; FXR: farnesoid X receptor; HA: hyaluronic acid; KPV: lysine-proline-valine; PEST1: peptide transporter1; PGA: polyglutamic acid; TFR: transferrin receptor. The image was created with the use of Servier Medical Art modified templates, licensed under a Creative Common Attribution 3.0 Unported License (https://smart.servier.com, accessed on 19 February 2022).

**Figure 5 ijms-23-04339-f005:**
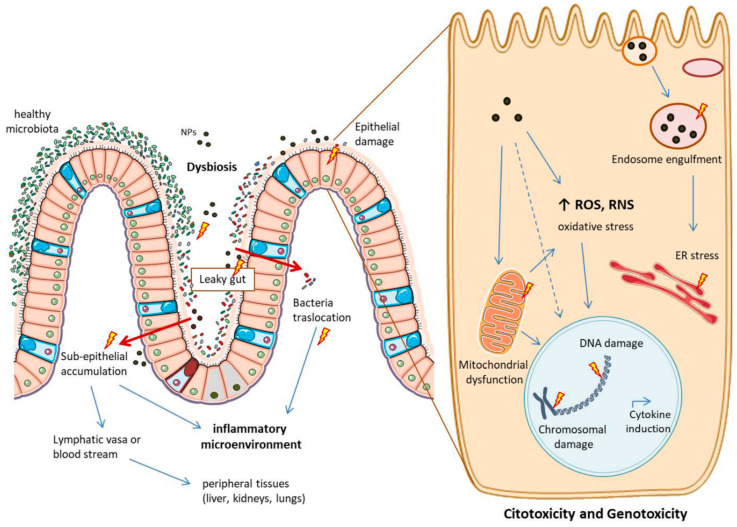
Main potential side effects of metallic NPs in the intestine and at cellular level. NPs can cause dysbiosis and tight junction rearrangements, favouring conditions of “leaky gut”. This can lead to bacteria translocation and NPs accumulation in the sub-epithelium, causing an inflammatory microenvironment. NPs in this space can pass into lymphatic vasa or into the blood stream, potentially reaching peripheral tissues. NPs can also directly harm the epithelium layer causing citotoxicity and genotoxicity once inside the cells. Here they can: accumulate into the endosomes, impair the vesicular trafficking and cause ER (endoplasmic reticulum) stress; increase the production of ROS and RNS, with oxidative stress and mitochondrial dysfunction; final outcomes can be cytokines induction and chromosomal and DNA damage. The image is original and was created with the use of Servier Medical Art modified templates, licensed under a Creative Common Attribution 3.0 Unported License (https://smart.servier.com, accessed on 19 February 2022).

**Table 1 ijms-23-04339-t001:** Main receptor-ligand interactions used for NPs functionalization in the intestine.

Reference	Receptor	Ligand	Cell Type Expression	Direct Penetration	Endocytosis
**Hua S 2020** [73]	Mannose Receptor	Mannose	Macrophages, Enterocytes, M cells	No	Yes
**Tian 2018** [74]	CD44	HA/CS	Macrophages, Intestinal Epithelial Cells	No	Yes
**Xiao, 2018** [75]	CD98	CD98 Fab’/single chain CD98 Ab	Intestinal Epithelial Cells, Macrophages	No	Yes
**Peng L, 2021** [76]	F4/80	F4/80 Ab Fab’	Macrophages	No	Yes
**Liu W, 2018** [77]	Macrophage Galactose Receptor	Lactobionic Acid	Macrophages	No	Yes
**Xi Z 2022, Álvarez-González, 2020** [78,79]	Folate Receptor	Folate	Macrophages, Epithelial Cancer Cells	No	Yes
**Yong, 2019** [80]	Transferrin Receptor	TFR Ab/Seven peptides	Intestinal Epithelial Cells	No	Yes
**Zhang W, 2021** [81]	PEST1	KPV	Macrophages, Intestinal Epithelial Cells	No	Yes
**Liu L, 2018** [82]	Mannose Receptor	TAT	Intestinal Epithelial Cells, Macrophages	Yes	No
**Azevedo, 2020** [83]	FcRn IgG	Albumin	Intestinal Epithelial Cells	No	Yes
**Huang X, 2021** [84]	FXR	Deoxycolic Acid	Intestinal Epithelial Cells	No	Yes
**Urimi, 2019** [85]	Calcium Sensing Receptor	PGA	Intestinal Epithelial Cells	No	Yes

CS: chondroitin sulphate; FXR: farnesoid X receptor; HA: hyaluronic acid; KPV: lysine-proline-valine; PEST1: peptide transporter1; PGA: polyglutamic acid; TAT: Trans-Activator of Transcription; TFR: transferrin receptor.

**Table 2 ijms-23-04339-t002:** Summary of the different NPs and functionalization for the delivery of insulin.

Reference	Core of the NPs	Further Functionalization for Adhesion/Passage	Release Control	Reduces Glycaemia in Animal Model
**Li L 2017** [88]	Chitosan	**CPP**	n/a	Yes
**Wu J-Z 2017** [89]	diethylene glycol dimethacrylate	n/a	phenylboronic acid	Yes
**Alfatama 2018** [90]	Alginate/Chitosan	n/a	n/a	Yes
**Czuba 2018** [91]	PLGA	**SDS**	n/a	Yes
**Fan 2018** [92]	Chitosan	**Deoxycholic acid**	n/a	Yes
**Hou 2018** [93]	Mesoporous silica nanoparticle	n/a	phenylboronic acid	Yes
**Jamshidi 2018** [94]	Chitosan	n/a	n/a	Yes
**Ji N 2018** [95]	Zein + CSA	n/a	n/a	n/a
**Liu L 2018** [82]	Chitosan + hydrogel	n/a	n/a	Yes
**Song M 2018** [96]	Cyclodextrin/chitosan	n/a	n/a	Yes
**Tian 2018** [74]	Chitosan/hyaluronic acid	n/a	n/a	Yes
**Wang W 2018** [97]	Polyamidoamine/polyaspartic acid/phenylboronic acid/PEG	**PEG**	phenylboronic acid	Yes
**Xu Y 2018** [98]	solid lipid nanoparticle + endosomal escape agent	n/a	n/a	Yes
**Zhang Y 2018** [99]	hydroxyapatite	**PEG**	n/a	Yes
**Zhang L. 2018** [100]	PLGA + chitosan + alginate	n/a	pH dependent	Yes
**Alsulays 2019** [101]	Solid lipid nanoparticles	**CPP**	n/a	Yes
**Guo 2019** [102]	Chitosan	**CPP**	n/a	yes
**Hu 2019** [103]	phospholipids	n/a	n/a	Yes
**Jamwal 2019** [104]	dextran	n/a	Glucose oxidase	n/a
**Ji 2019** [105]	Chitosan/zein-carboxymethylated short-chain amylose	n/a	n/a	Yes
**Mohammadpour 2019** [106]	PLGA + chitosan	n/a	Glucose oxidase	Yes
**Muntoni 2019** [107]	Lipid nanoparticles	n/a	n/a	Yes
**Mudassir 2019** [108]	Methyl methacrylate/itaconic acid nanogels	n/a	pH dependent	Yes
**Tsai 2019** [109]	Chitosan + fucoidan	n/a	pH dependent	n/a
**Urimi 2019** [85]	Chitosan	**Polyglutamic acid**	n/a	Yes
**Azevedo 2020** [83]	Albumin	n/a	n/a	Yes
**Bai 2020** [110]	PLGA + glutamic acid conjugated amphiphilic dendrimer	n/a	n/a	Yes
**Chai 2020** [111]	Poly (acrylamido phenylboronic acid)/sodium alginate	n/a	Cicloborate (Glucose sensing) and glucose oxidase	Yes
**Chen Z 2020** [112]	Chitosan/Hyaluronic acid	**CPP**	n/a	Yes
**Cheng 2020** [113]	Poly (n-butylcyanoacrylate)	**Ratio insulin/Poly (n-butylcyanoacrylate)**	Ratio insulin/Poly (n-butylcyanoacrylate)	Yes
**Ding 2020** [114]	amphiphilic cholesterol- phosphate conjugate	n/a	pH dependent	Yes
**Han X 2020** [115]	Zwitterionic micelles	**Betaine**	n/a	Yes
**Jana 2020** [116]	hyaluronic acid	n/a	Glucose oxidase	n/a
**Mumuni 2020** [117]	Chitosan/mucin	n/a	n/a	yes
**Sladek 2020** [118]	Hyaluronic acid/chitosan	**Sucrose laurate**	n/a	Yes
**Sudhakar 2020** [119]	Chitosan	n/a	pH dependent	Yes
**Tan X 2020** [120]	Mesoporous silica	**PEG + CPP**	n/a	Yes
**Wang T 2020** [121]	Lipid nanoparticles	n/a	n/a	Yes
**Zhou S 2020** [122]	Chitosan	**PC6**	pH dependent	Yes
**Zhou X 2020** [123]	Alginate	n/a	Glucose oxidase	Yes
**Zhou Y 2020** [124]	FeCl_3_·6H_2_O + BTC	**SDS**	pH dependent	Yes
**Bao X 2021** [125]	Zein/casein-dextran	**Cholic acid**	n/a	Yes
**Benyettou 2021** [126]	Nanoscale imine-linked covalent organic frameworks	n/a	pH dependent	Yes
**Cui 2021** [127]	Chitosan + Hyaluronic acid	**Biotin**	n/a	Yes
**Huang X 2021** [84]	layered double hydroxide nanoparticle + hyaluronic acid	**Deoxycholic acid**	n/a	Yes
**Kim WJ 2021** [128]	POSS-APBA	n/a	phenylboronic acid	n/a
**Li H 2021** [129]	polyphosphoesters-based copolymer	n/a	phenylboronic acid	Yes
**Li J 2021** [130]	Alginate/chitosan	n/a	pH dependent	Yes
**Liu X 2021** [131]	PLGA/PEG	**Angiopep-2**	n/a	Yes
**Qin 2021** [132]	Mesoporous silica + Alginate + Boronic acid Mesoporous silica + Chitosan + boronic acid	n/a	phenylboronic acid	Yes
**Rao 2021** [133]	Porous silicon nanoparticles	**Zwitterionic dodecyl sulfobetaine**	n/a	Yes
**Volpatti 2021** [134]	Polycation	n/a	Glucose oxidase	Yes
**Wang W 2021** [135]	PLGA	**Chitosan + Cholanic acid**	n/a	Yes
**Zhang Y 2021** [136]	mesoporous silica nanoparticles	**CPP**	n/a	Yes
**Fu 2022** [137]	Glycopolymer	n/a	phenylboronic acid	Yes
**Li J 2022** [138]	PLGA-Hyd-PEG	**PEG**	n/a	Yes
**Martins 2022** [139]	Lignin-encapsulated silicon	**Fc fragment of IgG**	pH dependent	n/a
**Reboredo 2022** [140]	Zein	**PEG**	n/a	Yes
**Rohra 2022** [141]	Gold nanoparticle-encapsulated zeolitic imidazolate framework-8	n/a	Glucose oxidase	n/a
**Xi Z 2022** [78]	PLGA/PEG	**PEG, folate and charge-convertible tripeptide**	n/a	Yes
**Xu 2022** [142]	konjac glucomannan/concanavalin A	n/a	Glucose sensing	Yes

Most recent articles were considered (starting from 2017). APBA: 3-Aminophenylboronic acid monohydrate; BTC; 1,3,5-Benzenetricarboxylic acid; CPP: cell-penetrating peptides; CSA: Carboxymethylated Short-Chain Amylose; PC6: poly(acrylic acid)−cysteine−6-mercaptonicotinic acid; PLGA: poly (d, l-lactic-co-glycolic acid); POSS: PSS-[2-(3,4-epoxycyclohexyl)ethyl]-heptaisobutyl substituted.

**Table 3 ijms-23-04339-t003:** List of new selected insulin-based nanomedicines in completed trials in recent years.

Drug Name	Company	Material Used	Delivery Route	NCT Number	Outcome
**HDV-I**	Diasome Pharmaceuticals and Integrium	Liposomal bilayer containing Hepatic directed vesicles (HDV)–insulin	Oral	NCT00814294 NCT00521378	No results posted No results posted
**Oshadi**	Oshadi Drug Administration Ltd.	Silica-based NP with polysaccharides and oil combination of insulin, proinsulin, and C-peptide	Oral	NCT01120912 NCT01973920 NCT01772251	No results posted No results posted No results posted
**ORMD-0801**	Oramed Pharmaceuticals Ltd.	Human recombinant insulin contained in an enteric coated capsule with adjuvants	Oral	NCT02496000	Positive: ORMD-0801 was well tolerated and had a significant anti-hyperglycaemic effect, not associated with any serious hypoglycaemia conditions.
				NCT03467932	No result posted
				NCT00867594	No results posted
**IN-105**	Biocon Ltd.	PEGylated-Tregopil (modified form of human insulin)	Oral	NCT01035801	No results posted
				NCT04141423	No results posted
				NCT03430856	Positive: IN-105 is relatively well tolerated as compared to Insulin Aspart.
**GIPET^®^ (insulin 338)**	Novo Nordisk	Micelles-loaded human recombinant insulin	Oral	NCT01931137 NCT02470039 NCT02304627 NCT01809184 NCT01796366	No results posted No results posted No results posted No results posted No results posted
**Nasulin^TM^**	CPEX Pharmaceuticals	Formulation of CPE-215 (cyclopentadecalactone) recombinant human insulin	Intranasal	NCT00850096	Positive: Nasulin^TM^ is relatively well tolerated and increased the absorption of insulin with repeated dosing on the same nostril.
**Afrezza1**	Mannkind and Sanofi	Technosphere microparticles (fumaryl diketopiperazine (FDKP))of recombinant human insulin	Inhaled	NCT03143816 NCT02485327	Positive: Afrezza1 improves post-prandial glucose without increasing hypoglycaemia. No results posted

Data obtained from www.clinicaltrials.gov accessed on 4 April 2022.

**Table 4 ijms-23-04339-t004:** Summary of the main food additives containing NPs used in food industry at the moment, their current regulations and estimated intakes according to EFSA and the main concerns raised in literature for their use.

Food Additive	Main Content	Properties	Current Recommendations (EFSA)	Estimated Intake in Toddlers	Estimated Intake in Children	Estimated Intake in Adults	Main Concerns
**E171** [37]	TiO_2_	Food colourant (white)	No longer considered safe when used as food additive	0.9–12.8 mg/kg/day	1.9–11.5 mg/kg/day	0.7–6.7 mg/kg/day	Genotoxicity. DNA damage. Accumulation in tissues. Inflammation, dysbiosis, leaky gut worsened in pre-existing conditions.
**E172** [192]	Fe oxides and hydroxides	Food colourant (yellow, red, black), Food supplements	No limitations at the moment. Need to distinguish the different compounds. Need for more studies on genotoxicity.	0.4–10.5 mg/kg/day	1.4–9.2 mg/kg/day	0.3–2.4 mg/kg/day	Genotoxicity. Limited studies at the time of assessment (2015).
**E173 *** [190]	Al	Food colourant (grey)	TWI of 1mg/kg/week	n.a	n.a.	0.2–1.5 mg/kg/week	Ions accumulation in tissues, including nervous system. Very few data at the moment.
**E174** [188]	Ag	Food colourant (silver-grey), antimicrobial agent	Need for more data on E174 characterisation. Need for specifications of the mean particle size distribution and NPs %.	0.003–0.08 mg/kg/day	0.01–0.11 mg/kg/day	0.001–0.03 mg/kg/day	Potential release of Ag ions. Potential: cytotoxicity, induction of oxidative stress, inflammatory response, dysbiosis.
**-** [187]	Ag	Antimicrobial agent in food packaging	Under the intended and tested condition of use do not give rise to toxicological concern.	<0.9 ug ion/kg/day (ADI)	<0.9 ug ion/kg/day (ADI)	<0.9 ug ion/kg/day (ADI)	Potential release of Ag ions.
**E175** [189]	Au	Food colorant (yellowish gold)	Need for specifications of the mean particle size distribution and NPs percentage. Currently not enough data.	0.01–0.26 μg/kg/day	0.04–0.33 μg/kg/day	0.01–0.09 μg/kg/day	Potential accumulation in tissues, but more data needed.
**-** [186]	ZnO	Antimicrobial agent and UV-light adsorber in food packaging	Migration only in the form of ions is observed, but lower than the specific migration limit → ok. Zinc upper limit 25mg/person/day	n.a.	n.a.	n.a.	Upper limit could be exceeded since Zn is present in more sources other than food packaging
**E551** [191]	SiO_2_	Texture-improving agent, flavour-carrying agent	More stringent limitations on metals present in E551 formulations. Need for more studies to define a proper ADI.	18.5–39.4 mg/kg/day	10.5–31.2 mg/kg/day	4.9–13.2 mg/kg/day	Potential presence of metal contaminants.

Estimated intakes refer to the mean level of assumption according to the maximum level scenario in different European states expressed in mg/kg of body weight per day, unless otherwise stated. TWI= tolerable weekly intake; ADI = acceptable daily intake. * E173 is reported here for completeness of information, but EFSA recommendations regard Al in food, referring only to “aluminium” in general, without considerations on NPs presence. The latest update refers back to 2011.

## Data Availability

Not applicable.

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
