# Peer review of "Interactions between Nanoparticles and Intestine"

_ijms, 2022, doi:10.3390/ijms23084339_

Round 1

Reviewer 1 Report

The manuscript titled “Interaction between nanoparticles and intestine” by Vitulo et al. has been reviewed where the authors have intended to provide an overview on the fate of nanoparticles in the intestine. The work is interesting to read and informative. In my opinion there are not many reviews published so far to explore and understand this issue, hence this work is timely. However, there are a few comments appended below, which authors might like to consider making it more informative before it could be considered for publication.

  1. How much work has been done in the field of designing nanoparticles for drug delivery through the oral route, is not so clear in this review? I would like to suggest providing some discussion in this line and to present a figure showing how this field is progressing (how many articles have been published per year for a certain period of time). This will also help to predict the direction this field is moving.
  2. It would be good to include one more figure showing different routes of drug delivery (cf. Adv. Sci. 2022, 2105373, DOI: 10.1002/advs.202105373) and the different types of nanoparticles use specifically in oral routes.
  3. In Table 1, authors have listed different types of interactions between nanoparticles and intestine. Is it possible to present different types of interactions through a summary figure so that the authors can easily learn how these interactions are occurring?
  4. It seems the authors did not pay enough attention during the preparation of the manuscript. The Table captions need to appear above each table.
  5. In section 3.1, Insert the full form instead of the abbreviated form IBD.
  6. Before the section 5 (Conclusion), it would be good to add a summary presenting two aspects – (i) Advantages of using nanoparticles in oral administration and (ii) the unavoidable impact of toxicity.
  7. Based on these understanding, the future perspective should also be presented in the conclusion (Future Perspective and Conclusion) where the authors are encouraged to provide their expert opinion on the future direction of nanoparticles, which properties are to incorporate into the nanoparticles so that the NPs can be of better use for oral delivery.
  8. Authors have not commented/summarized on the possibilities of these nanoparticles to be used in clinical trials/practices. What is the status?
  9. Authors need to incorporate some English corrections in various places throughout the manuscript.

Author Response

 Reviewer 1
We do thank the reviewer for the comments, which helped us to improve the overall quality of the manuscript.
A point-to-point response is following below. Please note that the line numbers reported for the corrections
refer to the version showing all the revisions, as suggested by the journal.
1. How much work has been done in the field of designing nanoparticles for drug delivery through the oral
route, is not so clear in this review? I would like to suggest providing some discussion in this line and to
present a figure showing how this field is progressing (how many articles have been published per year for
a certain period of time). This will also help to predict the direction this field is moving.
A figure (Fig.1) representative of the growth of the published literature has now been included, along with a brief comment in the introduction (lines 38-48).
2. It would be good to include one more figure showing different routes of drug delivery (
cf. Adv. Sci. 2022,
2105373, DOI: 10.1002/advs.202105373) and the different types of nanoparticles use specifically in oral
routes.
The former Fig.1 (now Fig.2) has been modified including the different types of nanoparticles and delivery route in panel a (Lines 72-78).
3. In Table 1, authors have listed different types of interactions between nanoparticles and intestine. Is it
possible to present different types of interactions through a summary figure so that the authors can easily
learn how these interactions are occurring?
A graphical representation of the different types of interaction between nanoparticles and intestine reported in table 1 has now been included as Fig.4 (lines 466-472).
4. It seems the authors did not pay enough attention during the preparation of the manuscript. The Table
captions need to appear above each table.
We do apoligize for the mistake. The table captions have now been moved above each table, whereas the footnotes are below.
5. In section 3.1, Insert the full form instead of the abbreviated form IBD.
The abbreviated form IBD has now been substituted by the full form “Inflammatory Bowel Disease”.
6. Before the section 5 (Conclusion), it would be good to add a summary presenting two aspects – (i)
Advantages of using nanoparticles in oral administration and (ii) the unavoidable impact of toxicity.
As suggested, a summary section (section 5) has now been added to the manuscript (lines 1033-1063).
7. Based on these understanding, the future perspective should also be presented in the conclusion (Future
Perspective and Conclusion) where the authors are encouraged to provide their expert opinion on the future
direction of nanoparticles, which properties are to incorporate into the nanoparticles so that the NPs can be
of better use for oral delivery.
The conclusion section has been expanded, and now includes some considerations regarding NPs properties and their capacity to provide a better oral delivery (lines 1076-1098, 1101-1103).

8. Authors have not commented/summarized on the possibilities of these nanoparticles to be used in clinical
trials/practices. What is the status?
A brief comment on the status of clinical trials regarding the use of NPs for insulin delivery has been added in section 4 (lines 1115-1161). A table summarizing these completed trials has also been included (Table 3).
Furthermore, a comprehensive table of the currently approved NPs for therapeutic use, as reported on
www.clinicaltrials.gov, has been inserted in supplementary material (Table s1).
9. Authors need to incorporate some English corrections in various places throughout the manuscript.
The manuscript has been revised, corrections have been incorporated in the manuscrip

Reviewer 2 Report

  1. In this manuscript, Barisani et al summarized the interactions between nanoparticle and intestine. This work is interesting. In my opinion, this manuscript is suitable for publication in IJMS in future.
  2. In page 1 line20, it is better to give the full name of E171 and E174.
  3. NPs should be unified in full text. In the manuscript, NPs were wrote as Nps or NP, etc. 
  4. The interaction between NPs and gut microbiota should be included,especially, the adverse effect.
  5. In my opinion, the nanozyme for the treatment of Inflammatory bowel diseases should be added in section 3.
  6. In page 15 line 538, the sentence should revised.
  7. Besides, some format should be improved, for example, Fe3O4, TNFα etc.
  8. In conclusion section, based on this field, please provide possible directions for future material design.  

Author Response

 Reviewer 2
We do thank the reviewer for the comments, which helped us to improve the overall quality of the manuscript.
A point-to-point response is following below. Please note that the line numbers reported for the corrections
refer to the version showing all the revisions, as suggested by the journal.
1. In page 1 line20, it is better to give the full name of E171 and E174.
As suggested, the food additives codes “E171” and “E174” are now explained in the abstract.
2. NPs should be unified in full text. In the manuscript, NPs were wrote as Nps or NP, etc.
We do apologize for the mistake. Now the abbreviation “NPs” has been unified throughout the text and
figures.
3. The interaction between NPs and gut microbiota should be included,especially, the adverse effect.
We appreciated the suggestion. A paragraph on the interaction between microbiota and NPs is now present
in section 1.2 (lines 154-246). Due to the vastity of the subject it cannot cover every single aspect, but we do
hope that it is enough to be food for thought for the readers.
4. In my opinion, the nanozyme for the treatment of Inflammatory bowel diseases should be added in section
3.
As suggested, the nanozyme for the treatment of Inflammatory bowel diseases has been inserted in the
nanoparticle loading drugs section (lines 684-690)
5. In page 15 line 538, the sentence should revised.
As requested, this sentence has been revised (lines 889-891)
6. Besides, some format should be improved, for example, Fe3O4, TNFα etc.
We have revised the manuscript and verified/improved the format of the various cited molecules.
7. In conclusion section, based on this field, please provide possible directions for future material design.
The conclusion section has been expanded, and now includes some considerations regarding NPs properties
and their capacity to provide a better oral delivery (lines 1076-1098, 1101-1103).

Round 2

Reviewer 1 Report

The authors have substantially improved the manuscript and replied satisfactorily to my concerns raised earlier. It gives me pleasure to recommend this work for publication in IJMS. A small concern is that in Figure 2a, the figure of "Polymeric Nanoparticle" might have taken from Biorender (https://biorender.com/icon/chemistry/nanoparticles/polymeric-nanoparticle-2-colors/), I am not sure though. Authors and the editorial office should make sure that there is no copyright issue.

Reviewer 2 Report

After revision,  this manuscript can be accepted for publication in my opinion.